

**Assessment of Physiological Stress and Bark Beetle-Induced**
**Mortality in Fir Trees, Zao Mountains, Japan**
Anna Trigubenko [a], Maximo Larry Lopez Caceres [a], Hisaya Shimizu [a], Tatiana A. Shestakova [b,c],
Vladislav Bukin [a], Noboru Kojima [a]
[a] Faculty of Agriculture, Yamagata University, Tsuruoka, Japan;
[b] Department of Agricultural and Forest Sciences and Engineering, University of Lleida, Lleida,
Spain;
[c] Joint Research Unit CTFC–AGROTECNIO–CERCA, Lleida, Spain.
Correspondence to: Anna Trigubenko (annya.tri.99@gmail.com), Maximo Larry Lopez Caceres
(larry@tds1.tr.yamagata-u.ac.jp)
**Abstract**
In the Northern Hemisphere, bark beetles are responsible for high tree mortality rates in
forest ecosystems. In recent decades, forest pest outbreaks have increased in frequency and scale
related to climate change. Although, many studies have focused on the effect of pest outbreak on
forests, there are few studies focusing on the early physiological stress on trees preceding the
infestations. In the treeline of Zao Mountains in northeastern Japan, a double pest infestation of
totrix moth (*Epitonia piceae*) and bark beetles (*Polygraphus Proximus*) that caused severe tree
mortality in a natural fir forest (*Abies mariesii*), which is the first reported case worldwide of



treeline retreat caused by bark beetle infestation. In order to understand forest dynamics prior to
the outbreak, tree rings samples were collected from 20 trees in the treeline (dead trees) and 40
trees from living trees (healthy and damaged) at lower altitudes. In these samples a
dendrochronological and carbon stable isotope analysis ($\Delta^{13}$C) was performed. Results indicate a
declining trend in tree-ring indices (TRI) for dead trees, while living trees showed a strong an
annual fluctuation, but did not show any declining trend. Healthy and damaged trees maintained
relatively stable $\Delta^{13}$C values (14.9‰ - 18.5‰), reflecting stable physiological activity even in the
partially defoliated damaged trees. During the years the infestation lasted, there was no response
from tree rings $\Delta^{13}$C (16.2‰ and 16.4‰) to its surrounding environment in trees prior to their
death. The decreasing trend of $\Delta^{13}$C values in tree rings prior to pest infestation in dead trees
indicate a continuous decline in tree physiological activity caused by a tendency to close the
stomata due to environmental stress. In Zao Mountains evidence shows that extreme events in
winter lead to severe physical damage in trees, including fallen trees, caused by a combination of
heavy snow, strong winds and recently observed high snow density.  We speculate that this event
gradually weakened trees in the treeline. Another factor that is probably related to this trend is the
earlier snowmelt observed in the last two decades, which leads to decreases in soil moisture during
spring, when precipitation is the lowest. These findings suggest that $\Delta^{13}$C values in tree rings can
serve as early warning indicators of stress preceding severe natural disturbance and can contribute
to scientical based informing forest management strategies.

**1 Introduction**
Insect outbreaks appear to be increasing in frequency and intensity, a trend probably linked
to climate change (Agne et al., 2018; Jactel et al., 2019; Przepiora et al., 2020). Globally, insect-
disturbed forests span 85.5 million hectares, representing 3% of the total forested area (2807
million hectares) across boreal, temperate, and tropical regions (van Lierop et al., 2015). This
alarming trend underscores the need for improved understanding of the drivers and impacts of



such disturbances. Bark beetle-induced mass forest mortality has become a large-scale
destabilizing factor for forest ecosystems worldwide in recent decades (Cole and Amman, 1980;
Pavlov et al., 2020).
Among insect species responsible for widespread damage, *Polygraphus proximus*, a non-
aggressive phloephagous bark beetle, has garnered significant attention due to its detrimental
effect on environmentally disturbed forests. Native to northeastern China, Korea, Japan, and the
southern part of the Russian Far East (Nobuchi, 1966; Koizumi, 1977; Kerchev, 2014), this insect
has invaded Western Siberia and European Russia, where it has devastated vast areas of fir forests
over the past decade (Kononov et al., 2016; Kharuk et al., 2019). In Japan, *Polygraphus proximus*
has caused extensive damage to Abies species (Tokuda et al., 2008; Takagi et al., 2018, 2021;
Chiba et al., 2020), threatening the ecological stability of these forests.
As with other bark beetles, once *Polygraphus proximus* populations reach outbreak levels,
they are capable of infesting even healthy trees. Such outbreaks, similar to those caused by the
mountain pine beetle (*Dendroctonus ponderosae*), have led to the mortality of seemingly healthy
trees across millions of hectares. In Zao Mountains, a large-scale bark beetle outbreak between
2012 and 2016 resulted in the devastation of pristine *Abies mariesii* forests over hundreds of
hectares, especially those close to the treeline. This outbreak has drastically altered the landscape
and is expected to have long-term ecological consequences in the region. Bark beetle infestations
not only reduce timber production and quality but also disrupt nutrient cycling, carbon uptake, and
ecosystem biodiversity, highlighting the far-reaching impacts of these disturbances. Bark beetle-
induced tree mortality also reduces the recreational and economic value of forests, affecting human
health, tourism, and local livelihoods. The study of the dynamics of forest degradation due to insect
outbreaks is of paramount importance, especially under climate change. As the affected area is
within a national park, the use of chemicals or other potentially harmful materials is prohibited.
Thus, it is necessary to identify early warning signals of physiological stress in trees, which could
enable proactive measures to be taken to prevent widespread infestation and mitigate its impact.



Forecasting tree mortality remains one of the most uncertain aspects of dynamic vegetation models
(Bugmann et al., 2019), especially when multiple disturbance factors interact. Identifying early
reliable indicators of physiological stress that are informative of, for example, the carbon-water
balance of trees is critical for improving predictive capabilities and forest management strategies.
Tree rings, which serve as natural archives, capture annual-resolution information of radial
growth and the environmental conditions at the time of tree ring formation. Among the most
important tools used in conjunction with dendrochronology, stable carbon isotope analysis of
individual rings stands as a powerful method for evaluating environmental influences on tree
physiology. Declining annual ring width, coupled with increasing carbon isotope ratios ($\delta^{13}$C), has
been correlated with reduced photosynthetic activity and stomatal closure, preceding tree mortality
by several years (Gessler et al., 2002; Cailleret et al., 2017). These patterns underscore the value
of tree-ring analysis and stable isotope studies in unraveling the complex interplay of climatic and
biotic factors influencing forest dynamics. Long-term decreases in annual ring width coupled with
increasing trends in $\delta^{13}$C can be used as early warning signals of physiological stress and recovery
of trees when subjected to severe disturbances (Lopez et al., 2018). Advanced dendrochronological
and isotopic approaches are key to understanding the interactions between physiological stress,
climatic variables, and bark beetle activity. By combining detailed growth and isotopic data with
regional climate records, predictive models can be developed to identify high-risk areas and inform
adaptive forest management practices (McCarroll and Loader, 2004; Bugmann et al., 2019).
The host species in this study, *Abies mariesii*, commonly known locally as Ooshirabiso, is
an evergreen tree native to subalpine regions of Japan, which is usually found at elevations ranging
from 1300 to 2900 meters. Fir forest thrives in cold temperate rainforests characterized by high
rainfall and heavy snow (Tanaka and Matsui, 2007). These trees, which typically grow to heights
of 10 to 30 meters, play a critical role in maintaining ecosystem stability in Japan's mountainous
regions. However, the sudden loss of *Abies mariesii* due to bark beetle outbreaks has led to the
near-total collapse of forest ecosystems in the treeline (Chiba et al., 2017).



In this study, we hypothesize that the combined effects of environmental factors in Zao
Mountains in recent decades have affected stomatal conductance and intercellular $CO_2$
concentrations, leading to changes in wood isotopic signatures. Therefore, the specific objectives
of this study are:
1. To determine the tree growth pattern of fir trees three decades prior to pest infestation;
2. To compare the carbon isotope signatures (carbon isotope discrimination, $\Delta^{13}C$) of

healthy, damaged and dead trees (before death in 2016) and evaluate their

environmental control for each of the health categories from 1993 to 2022.


## 2 Methodology

### 2.1 Study site

Zao Mountains are located in an active volcanic range on Honshu Island, in northeastern
Japan on Honshu Island (38.14°N, 140.44°E), representing a region of ecological significance and
biodiversity. Spanning 11 hectares within Yamagata Prefecture, the study area is situated at the
border of Yamagata and Miyagi prefectures. The forest composition includes both pure stands of
Maries fir (*Abies mariesii*) at higher altitudes and broadleaf mixed forests interspersed with a few
deciduous and coniferous species such as *Fagus crenata, Acer tschonoskii,* and *Sorbus commixta*
in lower areas. The understory across the altitudinal gradient is dominated by Sasa grass, which
stabilizes soil and enhances nutrient cycling (Nguyen et al., 2021; Tran et al., 2024; Hu et al.,

2022).

The altitudinal gradient significantly influences forest structure and dynamics. At
elevations between 1468 m and 1535 m, tree density averages to 400 trees per hectare, with a
marked decline in tree size, height, and canopy area as altitude increases. The dominance of *Abies*
*mariesii*, which covers 87% of the forested area - is juxtaposed with patches of mixed coniferous



and deciduous vegetation. At higher elevations between 1508 m and 1714 m, close to the treeline,
over 92% of fir trees have succumbed to bark beetle (*Polygraphus Proximus)* infestation, resulting
in dramatic shifts in forest composition and structure (Fig. 1) (Saito and Chiba, 2017).

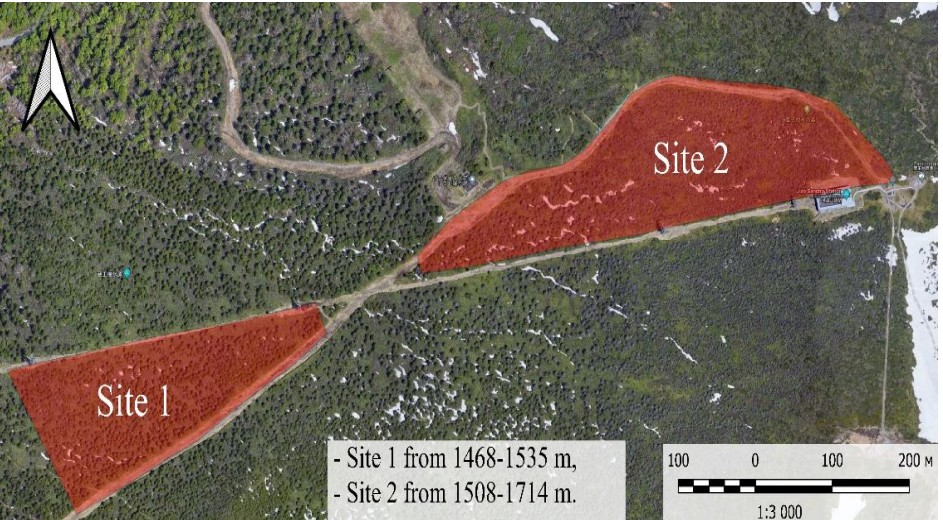

**Figure 1.** Study site located in Zao Mountains.

The ecological setting of Zao Mountains is further complicated by its status as an active

volcanic region. Frequent seismic activity, volcanic emissions, and extreme weather events such
as typhoons interact with biotic stressors, to shape forest health and resilience (Takagi et al., 2021).
These interactions create a dynamic environmental mosaic, making management of those forests
a challenging task. The forests of Zao Mountains also hold significant conservation value, serving
as carbon sinks and providing critical ecosystem services such as water regulation and biodiversity
maintenance. By synthesizing field observations, dendrochronological data, stable isotope records
and precise remote sensing information, this study uses an integrative approach which is critical
for developing predictive models and site-specific management strategies to mitigate future
impacts on subalpine forests in Japan.



**2.2 Tree core sampling and preparation**

Tree cores were collected in the summer of 2022. Two cores per tree were taken at breast height (1.3 m) from healthy, damaged, and dead individuals using a 5 mm increment borer (Haglöf, Sweden). One core was used for tree-ring dating, while the other was left intact for isotopic analysis. A total of 120 cores were collected, comprising 40 from healthy trees, 40 from damaged trees, and 40 from dead trees. This sampling strategy was designed to disentangle differences in growth and physiological responses between forest health conditions.

The preparation of tree-ring samples began with oven drying of the cores at 60°C for 48 hours. They were then glued into wooden holders. Hot glue was applied to hold the samples in place, ensuring stability during the subsequent processing steps. A polishing machine was then used to plane and smooth the top surface of the samples, removing irregularities and revealing the tree ring structure. This step improved the visibility of the annual rings for further analysis (Schweingruber, 1988; Stokes and Smiley, 1996).

**2.3 Tree-ring analysis**

High-resolution images (1000 dpi) of the polished cores were taken using specialized imaging systems. These images were used as the input for the WINDENDRO software (Regent Instruments Inc, Canada). Tree rings were visually cross-dated and measured to an accuracy of 0.01 mm. COFECHA was then used to validate the cross-dating of individual cores (Grissino-Mayer, 2001; Holmes, 1983). The software calculates the width of each annual ring, providing precise measurements essential for dendrochronological research.

COFECHA was used to validate the master chronology by cross-dating and detecting anomalies in tree-ring data, ensuring reliable $\delta^{13}C$ analysis (Grissino-Mayer, 2001; Holmes, 1983). Samples with high correlation with the Master chronology were selected for isotopic studies. ARSTAN standardized ring-width data, correcting growth anomalies linked to extreme climatic



events, enhancing the accuracy of environmental signal extraction (Cook & Krusic, 2005; Briffa
& Melvin, 2011).
**2.4 Stable isotope analysis**

Four cores that best correlated with the master chronology for each health status were

selected for isotopic analysis. Tree rings were manually separated using a fine scalpel under
magnification to ensure precise isolation of individual annual layers without cross-contamination.
Each ring was then finely cut into small pieces. Small pieces were ultrasonically cleaned in
distilled water to remove contaminants while preserving the cellulose structure. Rings
corresponding to the same year and health status were pooled into a single sample before analysis
(Leavitt 2008), except for every five years when rings were analyzed independently to estimate
the between-tree variability in carbon isotopes (Loader et al., 2013). Individual tree rings were
preserved for the years 1996, 2001, …, 2016 for dead trees and for the years 1997, 2002, …, 2022
for living trees. The resulting samples were homogenized using a ball mill. Approximately 0.6-0.8
mg of the dry material was weighed using a microbalance, packed in tin capsules, and analyzed
by isotope ratio mass spectrometry. Isotope ratios were expressed as per mil deviations using the
δ notation relative to Vienna Pee Dee Belemnite (VPDB). The accuracy of the analyses (SD of
standards) was 0.06‰.

To account for changes in $\delta^{13}C$ of atmospheric $CO_2$ ($\delta^{13}C_{air}$), we calculated carbon isotope

discrimination ($\Delta^{13}C$) from $\delta^{13}C_{air}$ and tree-ring $\delta^{13}C$ ($\delta^{13}C_{wood}$) following (Eq 1) (Farquhar et al.,
1982; Farquhar and Richards, 1984):
$$\Delta^{13}C = \frac{(\delta^{13}C_{air} - \delta^{13}C_{wood})}{1 + {\delta^{13}C_{wood}}/{1000}} \ (‰)$$   (1)
**2.5 Meteorological data**

Meteorological data for this study were collected at facilities of the Zao ropeway in Jizo

Mountain, located at 1676 m.a.s.l. The meteorological data include wind speed and direction, snow



depth, and temperature. Freeze-thaw cycles, high diurnal variability in air temperature, high
precipitation and strong western winds are the normal conditions in this region. Since
meteorological data from the Zao station is only available from 2012, a comparative analysis was
conducted using data from the nearby meteorological station at Yamagata city to extend the
temporal range. A comparative analysis over the common period (2012-2022) showed high
correlation between the monthly mean temperature of the two stations ($r$=0.99) (Fig. 2).
Consequently, the temperature data (mean, minimum, maximum) from the Yamagata station was
used as the primary source of climate information.

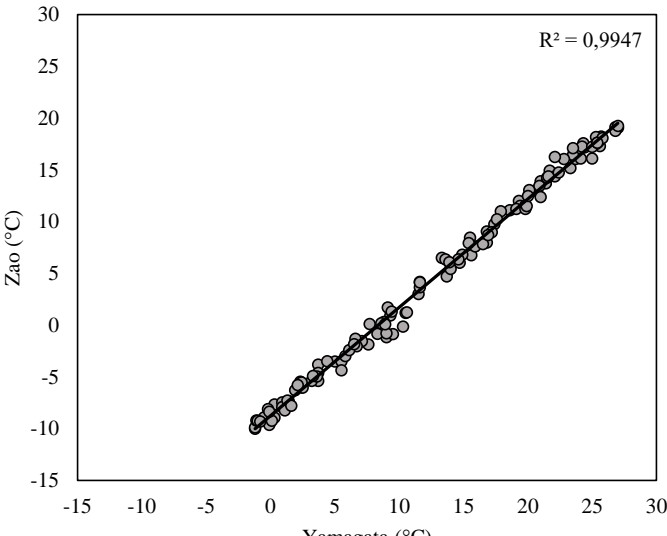


**Figure 2.** Correlation between monthly mean temperature records from the Zao ropeway facilities and

Yamagata meteorological station over the period 2012-2022.


**2.6 Unmanned Aerial Vehicles**





Orthomosaic imagery generated using UAV technology has proven particularly valuable
for documenting changes in crown conditions and forest health over time. High-resolution aerial
surveys from 2019 to 2022 revealed progressive damage (Fig. 3, Fig. 4), emphasizing the
importance of integrating spatial data with physiological analyses for comprehensive forest
monitoring. Advances in UAV-based monitoring provide an opportunity to enhance our
understanding of spatial and temporal patterns of forest degradation (Chiba et al., 2020; Tran et
al., 2024; Conciatori et al., 2024).

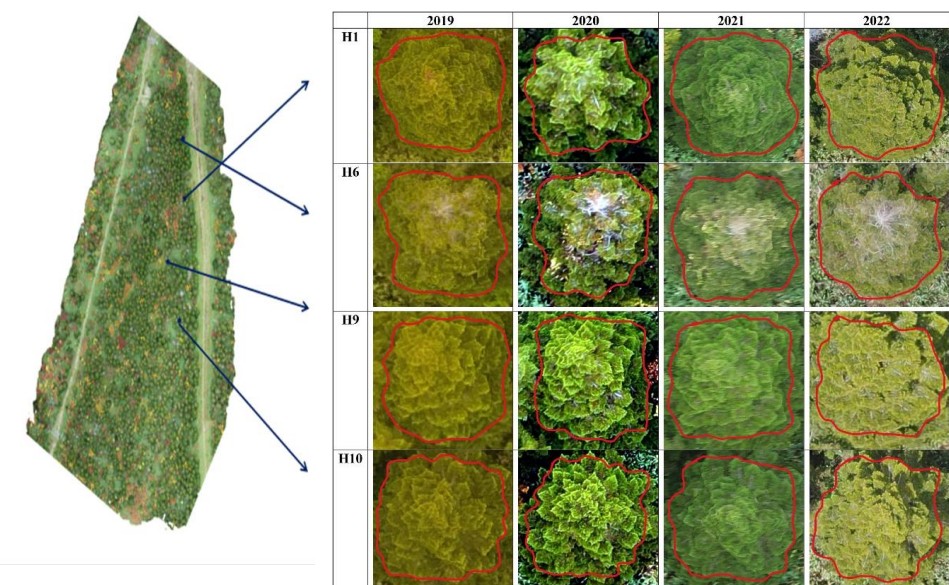

**Figure 3.** Chronological crown condition of some representative healthy (H) trees in orthomosaics for the

period 2019-2022.




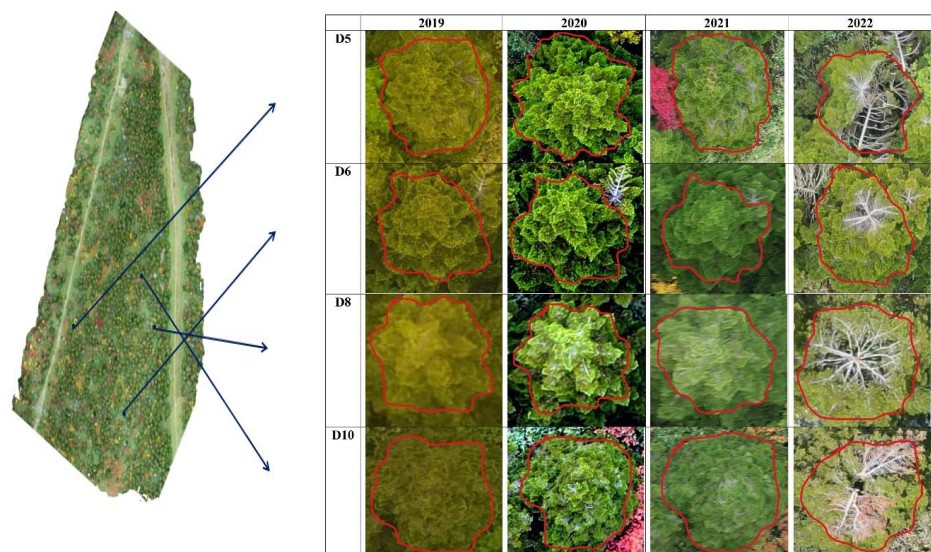

**Figure 4.** Chronological crown condition of some representative damaged (D) trees in orthomosaics for

period 2019-2022.

**3 Results**

**3.1 Tree-ring index**

The analysis of tree-ring indices (TRI) for healthy, damage, from 1993 to 2022 and for

dead trees 1993 to 2016 provides significant insights into tree growth dynamics and responses to

its environmental surroundings. The study period captured critical climatic and ecological

variations influencing forest health and resilience prior to pest infestation (Fig. 5).

Healthy and damaged trees were located at the same altitude and belonged to the same

forest area. Tree-ring growth represented by indexed tree-ring chronologies (TRI) for these two

health categories did not always have the same annual pattern, but most narrow and wide rings

were observed in the same years. The range of TRI values was similar for healthy trees (0.623-

1.300) and damaged trees (0.703-1.215), despite a difference in canopy defoliation observed

between the four trees classified as damaged and those with no defoliation, classified as healthy.



In the case of dead trees, they were located at higher altitudes with a different climate than that of
healthy and damaged trees, hence the different values observed in TRI during the period prior to
their death. For dead trees, the last year was determined based on the year when most of trees died
among the 20 samples.  TRI of healthy and damaged trees did not show a clear trend in the last 30
years. In contrast, dead trees showed a decreasing trend, especially from the year 1998, where a
clear shift to lower values was observed. Furthermore, the annual variability of tree-ring growth
was much lower over the past two decades, compared to the higher variability observed in the
1990s, or the higher variability observed for the TRI values of healthy and damaged trees.




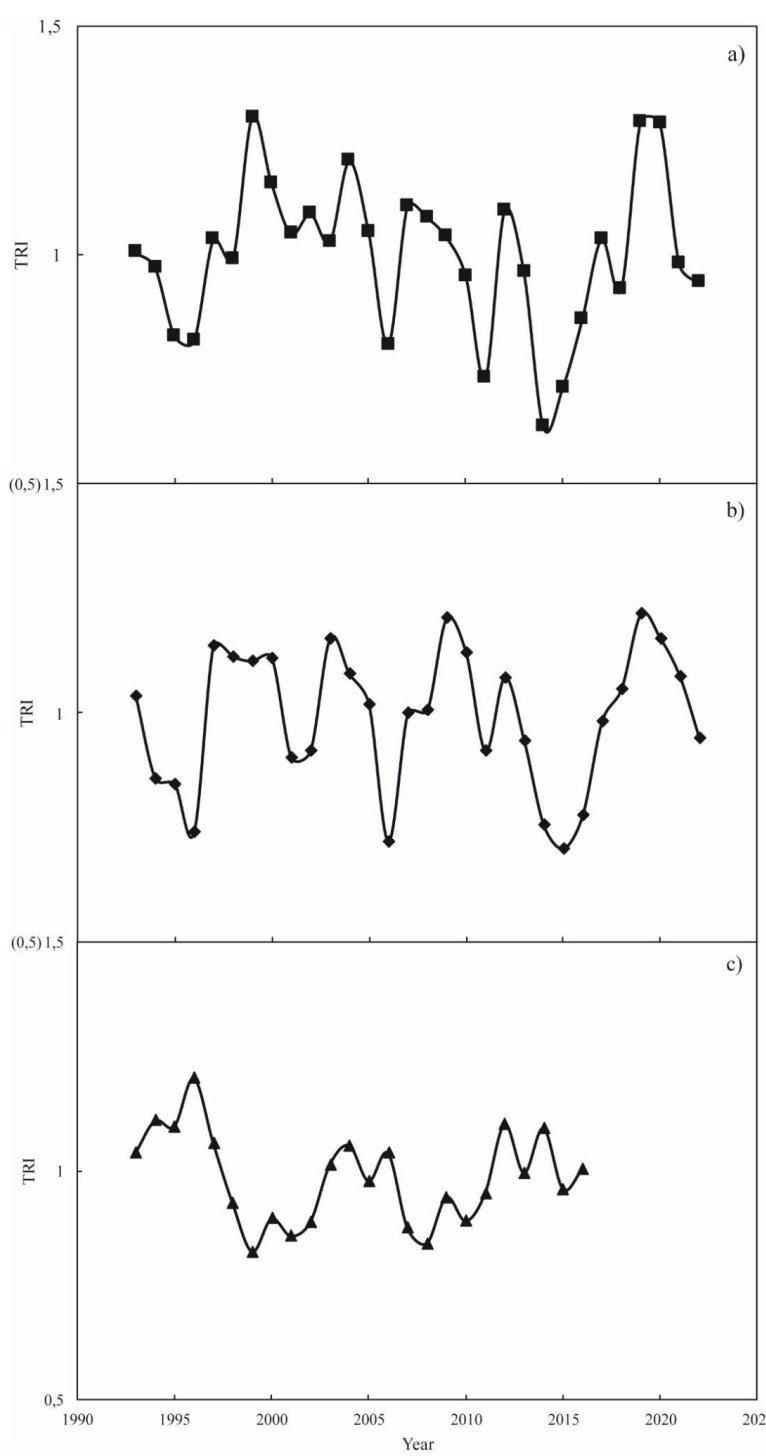


**Figure 5.** Trends in Tree Ring Index (TRI) from 1993-2022 for a) healthy, b) damaged, and c) dead trees.





**3.2 Carbon isotope discrimination (Δ¹³C)**

$\Delta^{13}C$ for the healthy trees remained relatively stable over the period 1993-2022 with values ranging from 14.9‰ to 18.5‰ with no clear trend for the last 30 years (Fig. 6a). In comparison, $\Delta^{13}C$ values for damaged trees showed a smaller range from 16.5‰ to 17.9‰ (Fig. 6 b). Healthy trees as well as damaged trees showed an increasing trend, especially from the beginning of the year 2000s, however the annual variability of $\Delta^{13}C$ values is smaller in damaged trees compared to the healthy ones. The composite oscillation with periodic individual sampling approach showed that the years chosen for multiple carbon isotope measurements represented well, the composite values as shown by the similar values found in the years when 4 individual samples were measured. There are some outliers as in 2018 in the healthy tree $\Delta^{13}C$ values but they show a steady trend.

In general, healthy and damaged trees did not show negative trends for the last 30 years, while $\Delta^{13}C$ values of dead trees for the period 1993-2016, showed a clear decreasing trend already from the start of the measuring period. The values oscillated within the range of 15.8‰ and 17.4‰ (Fig. 6c), with a clear annual variability for the period 1993- 2012, however, for the years 2012-2016, there is no response of $\Delta^{13}C$ to annual environmental variability, as the values remain the same for these years, which coincides with the pest infestation reported in this region.



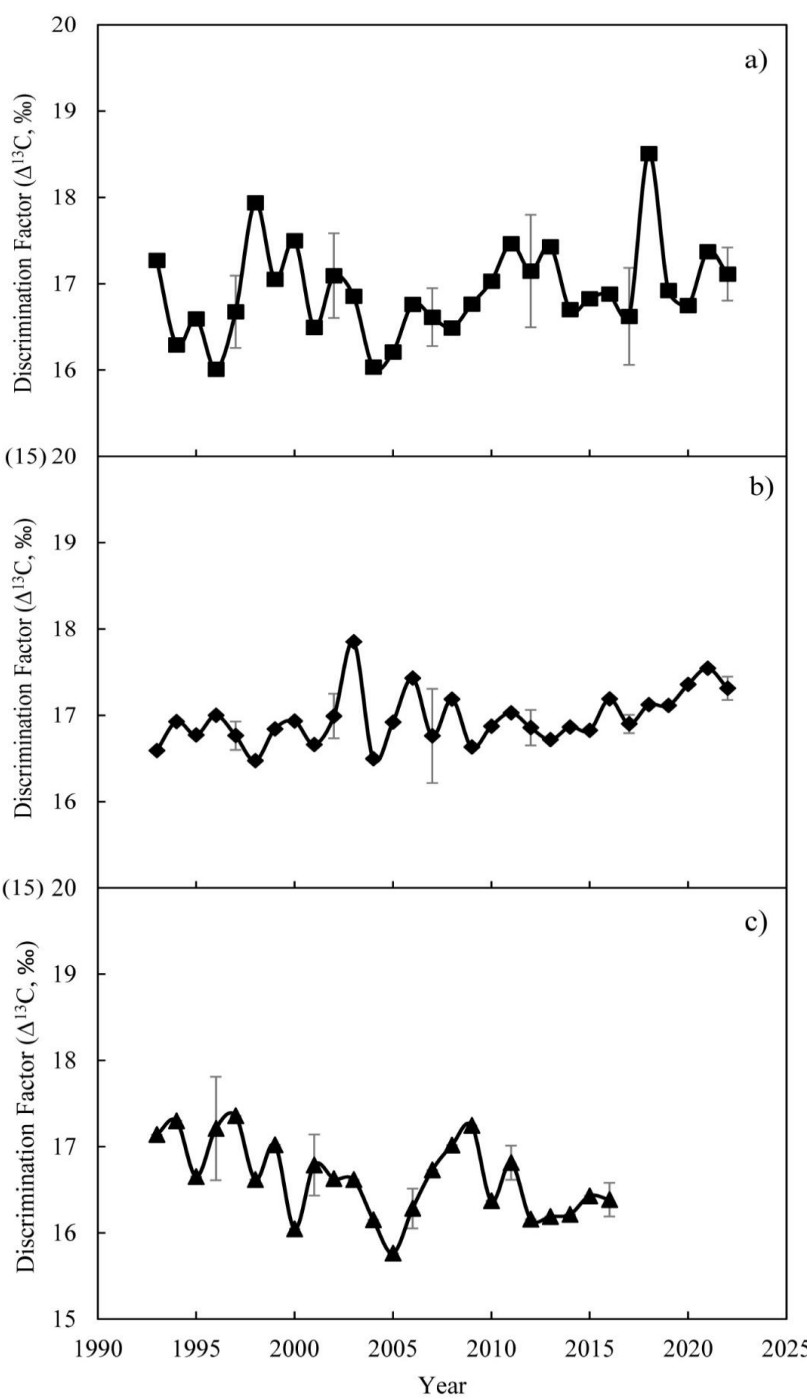




**Figure 6.** Carbon isotope discrimination ($\Delta^{13}C$) trends from 1993-2022 for a) healthy trees b) damaged
trees and c) dead trees. Error bars indicate the standard error of selected years for individually analyzed
trees.

**3.3 Environmental conditions**
Based on the strong correlation between air temperatures in Jizo Mountain and Yamagata
City, the annual average, minimum and maximum air temperatures from 1993 to 2022 data of
Yamagata city were examined (Fig. 7, a). Over the years, there is a clear upward trend in both
minimum and maximum temperatures, indicating a gradual warming. It appears that in the last 5
years, starting in 2018 to 2022, annual temperature remained the same with no clear annual
variability. Summer (June-August) vapor pressure deficit (VPD) data ranged from 0.5 to 0.9 from
1993 to 2022 (Fig. 7b). A peak was observed for the years 2010-2013, followed by a decline and
low annual variability in the last 5 years, with an average value for these years of 0.7 kPa.





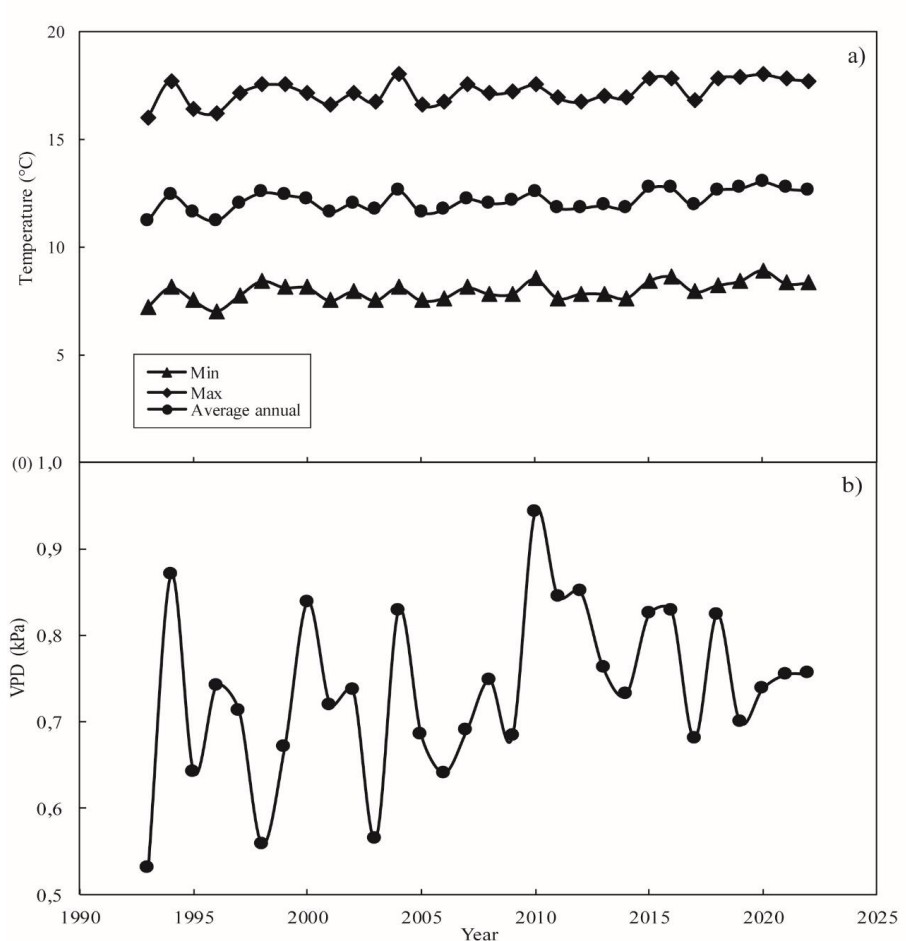

**Figure 7.** a) Average annual air temperature Trends in Yamagata (1993-2022); b) Summer (June-August) vapor pressure deficit (VPD) in Zao Mountain (1993-2022).

The analysis of snow depth data for Zao Mountain from 2012 to 2022 reveals significant seasonal and inter-annual variability (Fig. 8). The annual average snow depth ranged from 100 cm to 260 cm. The maximum value was observed in 2014 (260 cm) and the minimum during this period was reported in 2020 (100 cm). A peak after bark beetle infestation was recorded in 2018 (230 cm). From 2012 to 2020, a slight decreasing was observed, however, in the last three years



(2020-2022), a slight increase in snow depth has been observed, reflecting the unpredictability of
winter precipitation and accumulation patterns.

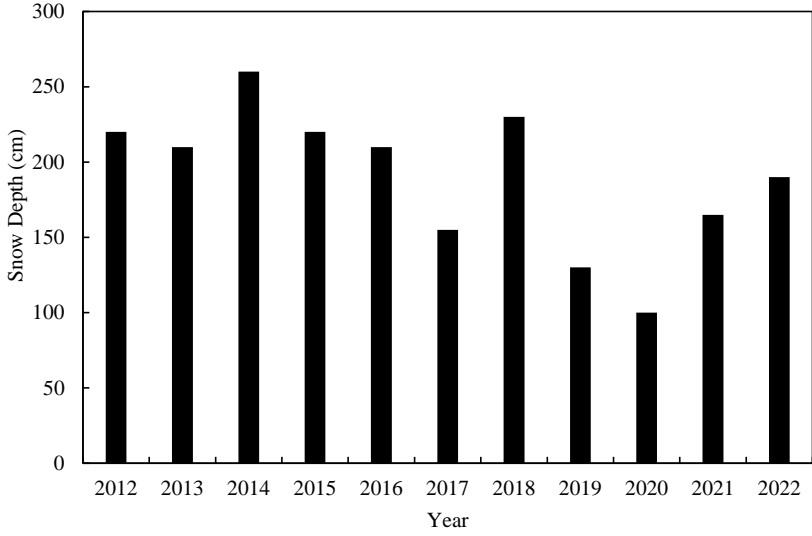


**Figure 8.** Annual Average Snow Depth in Zao Mountain (2012-2022).

Westerly winds dominated throughout the year, with particularly high wind activity during

the winter months (December to February). During this period, wind speeds frequently ranged
from 0 to 20 m/s, exceeding those recorded during the summer months. The west (W) direction
was dominant in the area, accounting for 63% of the total wind occurrences (Fig. 9). The southwest
(SW) direction is the second most frequent, contributing 12%, followed by northwest (NW) with
10%. Together, these three directions account for 85% of the total wind occurrences. Other
directions, such as south (S) with 7% and southeast (SE) with 3%, show limited influence, while
directions like north (N), east (E) and northeast (NE) collectively made up only 5% of the wind
direction patterns.





**Figure 9.** Wind direction distribution (2012-2022) in the Zao Mountains.

## 4 Discussion

The results of this study provide crucial insights into the difference in tree growth and physiological responses of *Abies mariesii* to environmental conditions prior to bark beetle infestations in the Zao Mountains. By analyzing annual ring width and stable isotope discrimination over the past three decades (period 1993-2016 for dead trees and 1993- for healthy and damaged trees), important signs of physiological stress preceding mortality were identified.

### 4.1 Tree ring growth and carbon isotope discrimination

Bark beetle infestations can lead to a decrease in tree-ring $\Delta^{13}C$ as plant carbon assimilation process is negatively affected. Results show that decreasing annual ring width coupled with decreasing carbon isotope discrimination ($\Delta^{13}C$) values can serve as early-warning signals of the long-term forest weakening that can lead to tree mortality when pest outbreaks occur. The decline of $\Delta^{13}C$ values in trees located in the treeline aligns with previous studies indicating that impaired stomatal conductance and reduced carbon assimilation reflect the effect of severe disturbances on trees (Lopez et al., 2018).



The trends in healthy tree TRI indicate that trees maintained relatively stable growth
patterns, while damaged trees exhibited smaller annual fluctuation in recent years, mainly caused
by their partial defoliation over the years. The difference between healthy and damaged TRI were
small, including higher and lower values during years of higher environmental variability. In
contrast, dead trees displayed a gradual decline in TRI values since the early 1990s, emphasizing
the cumulative effects of long-term stress which likely led to tree mortality. It is worth noticing
that healthy and damaged trees are located at a lower altitude compared to the dead trees growing
in the treeline, where trees are exposed to more extreme climate condition. As a result of the bark
beetle infestation the treeline in Zao Mountains has receded several hundred meters. Thus, the Zao
mountain this is a rare case of treeline recession as described in a meta-analysis by Harsch et al.
(2009), that reported that only 1% of all the treelines worldwide had receded vs 52% that had
advanced upwards.
The average age of trees in the two sites varies from 40-90 years old. Younger trees showed
higher ring growth related to higher photosynthesis rates (Ryan and Yoder, 1997) and higher cell
production (Rossi et al., 2008). They also use more topsoil water because of their widespread root
in the top soil (Børja et al., 2008), while old trees rely more heavily on the use of carbon reserves
of previous years, which can have higher $\delta^{13}C$ values (McCarroll et al., 2017; Timofeeva et al.,
2017), resulting in higher tree-ring $\delta^{13}C$ in old trees. Thus, lower soil moisture from snowmelt
could cause a major effect on younger trees, since older trees have deeper roots or can use more
reserves in Zao Mountains. However, the long-term stress observed in trees based on the
decreasing TRI and ($\Delta^{13}C$), suggests that the double infestation of *Epitonia piceae* and bark beetles
affected equally young and old trees, since tree mortality in this area reached 92%. When trees are
attacked by bark beetles, they experience physiological stress, which may affect their carbon
assimilation and allocation patterns. The $\delta^{13}C$ values in tree rings can reflect these changes.
However, the specific response of $\delta^{13}C$ to bark beetle attacks can vary depending on several



factors, including the severity of the attack, tree species, and local environmental conditions
(Ulrich et al., 2022; Kolb et al., 2019), as we have seen in our study.

Our results showed that the bark beetle attack reported in 2012 did not translate in a sharp

decrease of $\Delta^{13}C$ as it has been observed for severe disturbance (Lopez et al., 2018). Instead, a
small gradual decrease was observed but it was followed by lack of variation of TRI as well as
$\Delta^{13}C$, representing a loss of sensitivity of trees to their surrounding environment in the last 5 years
before death. The effect of *Epitonia piceae* attacks by itself does not necessarily lead to tree
mortality as it has been shown by Dulamsuren et al (2010) in larch trees after a severe gypsy moth
attack. However, the subsequent attack of bark beetle led to extended tree mortality.

Stress, such as the infestation of bark beetle in fir trees, can lead to partial closure of

stomata, reducing carbon dioxide uptake during photosynthesis. This can result in a decrease in
$\delta^{13}C$ values, as the plant discriminates less against $^{13}C$ under conditions of $^{12}C$ scarcity. Damaged
trees remained in the same condition for several years, but it is not clear if in the long term they
will recover to their original condition or if they will finally succumb to the infestation as it has
been already observed in trees in Zao Mountains.

The observed patterns confirm that dead trees experienced significant carbon assimilation

reduction before their demise, while healthy trees maintained stable physiological function.
Damaged trees showed some fluctuations in $\Delta^{13}C$ values, suggesting ongoing stress but not yet at
levels critical to mortality.

One of the reasons why trees in the treeline were attacked by bark beetles and subsequently

died was most probably the result of weakening by environmental factors. Other factor such as
root diseases, and competition for limited resources can increase stress among trees, could have
had an additional effect on increasing tree vulnerability. Trees at lower altitudes are taller and their
crowns are larger than in the treeline. Taller trees with more water availability, light, and nutrients
are more effective in repelling bark beetle attacks. Smaller vegetation, is the reflection of harsher





environmental conditions such as light, soil moisture, wind and snowfall. Milder environmental
conditions at lower altitudes provide trees with more suitable conditions to cope with pests or be
less susceptible to other disturbances. Rising temperatures and increasingly earlier snowmelt
periods can increase droughts especially during the spring period weakening tree defenses and
making them more susceptible to bark beetle attacks, although altitude played an important role in
the response of fir trees to bark beetle outbreaks as the same devastating effect was not observed
in lower altitudes.
**4.2 Role of Climatic Factors in Tree Vulnerability**

Due to the significant difference in elevation between the two sites, trees in the treeline

were exposed to more severe climate conditions, particularly strong wind and heavy snow in
winter. Meteorological analysis revealed that air temperatures have been increasing in recent
decades, which is leading to early snowmelt, depriving trees of soil moisture in spring. Tree
mortality stopped at a defined altitude delineating a clear line in the field by the year 2016, which
suggested the importance of elevation as a controlling factor but most importantly, that line appears
to represent the change in climate at the bottom of the treeline, at lower altitudes. One more factor,
that needs to be taken into account is the snow density, which in given years can cause an extra
physical stress on trees if combined with snow depth and usual strong winds increasing the
vulnerability of forests observed in the last decade. High wind speeds can increase the risk of
windthrow, where trees are uprooted or broken due to the combination of wind force and factors
such as soil conditions and root structure (Schindler et al., 2011). As it was observed in the winter
of 2021-2022, a large number of healthy trees fell down, usually broken at a height of 3 to 4 meters,
leaving fallen trees all over the forest in Site 1. These fallen trees, as it has been reported, can
intensify bark beetle reproduction in the disturbed forest (Louis et al., 2014, 2016), and could be
the trigger of further forest decline in Site 1 as it probably happened in Site 2. The recent increase
in VPD values, related to increases in air temperature, particularly in summer, may indicate further
drier air conditions. However, these values have been stable from the year 2011, which is also in



agreement with the stability of air temperature during this period. These trends suggest a potential
mitigation of extreme climatic conditions in Yamagata, at least for the last years. Despite the
physical damage that can be related to heavy snowfall, high snowfall accumulation can insulate
soil and tree roots, reducing freeze damage. The high variability of annual average snow depth
highlights the influence of larger-scale climatic factors, such as El Niño and La Niña, which can
impact snowfall intensity and duration (McClung, 2013).

**5 Conclusion**
This study showed that fir trees that died in the treeline of Zao Mountains as result of severe
bark beetle infestation, were in a continuous declining condition as the TRI and $\Delta^{13}C$ revealed.
From the beginning to end of the infestation (2012-2016), especially $\Delta^{13}C$ lost its sensitivity to
environmental factors, as its values remained stable during this period. In comparison, $\Delta^{13}C$ of
healthy and damaged trees, showed an increasing trend, representing better growth conditions and
a strong sensitivity to their surrounding environment. The results of this study suggest that carbon
stable isotopes can be used as an early warning system to evaluate the condition of vulnerable
forests such as those in the treeline. It also shows the first case reported worldwide of treeline
recession caused by bark beetle infestation under climate change.

**Data availability**
Data sharing is not applicable to this article.
**Sample availability**
Sample sharing is not applicable to this article.
**Author contribution**



A. Trigubenko: conceptualization, methodology, formal analysis, data curation, writing - original
draft preparation, visualization, laboratory analysis (stable isotope and dendrochronological
analysis), sample preparation, data interpretation.
M.L.L. Caceres: conceptualization, methodology, supervision, review and editing, project
administration, manuscript revision, critical input on research design, data validation.
H. Shimizu: meteorological data analysis, data curation, climate data integration, analysis of
environmental variables.
T.A. Shestakova: supervision, methodology, writing - review and editing, data validation, critical
review of manuscript, conceptual input, ethical and regulatory compliance.
V. Bukin: fieldwork coordination, sample collection, UAV data collection and analysis, map
creation, visualization, writing - review and editing, field data validation.
N. Kojima: laboratory assistance, sample preparation, technical support for stable isotope analysis,
data validation, writing - review and editing.
**Competing interests**
The authors declare that they have no conflict of interest.
**Acknowledgements**
We gratefully acknowledge the support provided by the members of the Smart Forest Laboratory
(Yamagata University) and to the laboratory of Prof. Matteo Garbarino from DISAFA department
(University of Torino) for data collection, sample preparation, and laboratory analysis (stable
isotope and dendrochronological analysis). Their contributions were crucial to the success of this
research.

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
