# Peer review of "Assessment of Physiological Stress and Bark Beetle-Induced"

_EGUsphere, 2025_

## Author Comment (AC1)

**Responses to Reviewer #1: RC1**

We sincerely appreciate your thorough evaluation and constructive feedback on our manuscript. Below, we provide detailed responses to all points raised. All comments have been carefully addressed and will be incorporated into the revised version of the manuscript.

On behalf of all co-authors,

Anna Trigubenko

**General comments:**

**Reviewer#1: The analysis lacks statistical support. See for example reporting of tree ring and 13c results.**

Response: We appreciate the reviewer's comment regarding the need for stronger statistical support. In the revised manuscript, we will include additional analyses to address this concern. Specifically, we will provide detailed geographical, structural and dendrochronological characteristics of the sampled sites, categorized by tree health status. We will also incorporate Basal Area Increment (BAI) calculations to better illustrate growth differences among healthy, damaged, and dead trees. Furthermore, we will present growth-climate and $\Delta^{13}$C-climate analyses, even in cases where correlations appear weak, in order to illustrate the potential influence of climate on tree physiology.

**Reviewer#1: The tree ring methods could be more clearly described. In particular, I am uncertain whether all «dead» tree cores contained the same years as it is not clear that all trees died in the same year.**

Response: We thank the reviewer for pointing out the need for clarification regarding the tree-ring methods. In the revised manuscript, we will carefully rework the sections describing the sampling and cross-dating procedures. While not all dead trees died in the same year, we note that, based on information provided by local forestry workers and national park staff, all trees classified as dead were already dead by 2016. For the stable carbon isotope analysis, we specifically selected trees whose cores included the final year of growth, ensuring comparability across individuals. We will revise the text to explain how we accounted for this variation, including the criteria used to determine the last year of growth for each dead tree and how these data were aligned for further analyses.

**Reviewer#1: Crossdating also was only from a single core per tree and the comparison among the dated core and the isotope core is not well described.**

Response: Thank you for your comment. The reviewer is correct that cross-dating was performed on a single core per tree, while the second core was kept intact for the stable isotope analysis. Please note that we sampled 20 trees per health category, which aligns with standard practice in dendrochronology. To minimize within-tree variability, both cores were extracted at the same height and orientation, whenever possible. Importantly: despite relying on a single dated core per tree, the resulting tree-ring chronologies were robust, with Expressed Population Signal (EPS) values exceeding 0.85 in all cases. We will clarify this methodological detail in the revised manuscript.

**Reviewer#1: I do not find the assessment of climatic drivers compelling, given no statistical correlation or regression analyses are described or reported.**

Response: Thank you for this insightful comment. We acknowledge the importance of statistically evaluating influence of climatic drivers on tree performance. In the revised manuscript, we will include the results of correlation analyses conducted between monthly climatic variables (mean air temperature and precipitation) and tree-ring width chronologies for each tree health category. These analyses will cover the period 1993-2022 for healthy and damaged trees and 1993-2016 for dead trees. Although the preliminary analyses show that the correlations are generally weak, we agree that reporting these findings adds transparency and supports our interpretation that climate is not a dominant driver of the observed tree mortality. We will revise the manuscript accordingly to reflect the outcome of the correlation analysis.

**Reviewer#1: Some justification for the use of bulk ring 13c instead of cellulose 13c should be included.**

Response: Thank you for your comment. We acknowledge the importance of selecting appropriate material for stable carbon isotope analysis. In our study, we chose to analyze $\delta^{13}C$ in bulk wood rather than extracting cellulose based on the review published by McCarroll and Loader (2004).

**Line comments:**

**Ln 53. Please provide citation/evidence the beetle is non-aggressive. I have seen suggestions of exactly the opposite (aggressive). https://research.fs.usda.gov/treesearch/37559**

Response: Thank you for pointing this out. We miswrote this point in the previous manuscript. Contrary to our earlier statement, the literature does describe the beetle (*Polygraphus proximus*)

as an aggressive species capable of mass-attacking healthy trees, particularly during outbreaks. We will quote it with the appropriate references, including the one you kindly provided.

**L61. This is kind of the definition of an aggressive bark beetle species, and D. ponderosae is an example of an aggressive species.**

Response: You are absolutely right. We appreciate the comparison with *Dendroctonus ponderosae*, which helps contextualize the behavior of *P. proximus* within the broader framework of bark beetle ecology.

**L96. Replace «thrives» with «is found»**

Response: Done.

**L98. Any examples of such critical roles?**

Response: We agree and will clarify this point as follows: In mountainous regions with intense rainfall (1,000 to 3,000 mm in Yamagata Prefecture), the root systems of *Abies mariesii* help anchor the soil, reducing the risk of landslides.

**L117. None of the three example species are coniferous**

Response: Thank for the suggestion. We will include appropriate examples in the revised manuscript.

**L137. Also looks like it's a ski area?**

Response: Yes, the study area is indeed part of a popular ski area. Zao Mountains, located within the Zao Quasi-National Park in northeastern Japan, are not only ecologically important but also serve as a major winter and summer tourism destination. The area hosts a large ski resort and is internationally recognized for the unique natural phenomenon known as «Snow Monsters» (Juhyo in Japanese), where *Abies mariesii* trees become completely covered with snow and ice due to strong winds. However, our sampling sites are located far from the ski facilities and are therefore not influenced by tourism-related activities.

**L145. So only a single core per tree was used for cross-dating? This is a bit unusual.**

Response: Thank you for your comment. Yes, we used a single core per tree for cross-dating, which is a common and widely accepted practice in dendrochronology, particularly when sample sizes are sufficient and cores are well-preserved. This approach is supported by standard dendrochronological protocols and has been shown to provide reliable results when cross-dating quality is high. We will clarify this point in the revised manuscript.

**L146. It looks like only four trees were eventually used per category? I understand limitations on 13c measurements, but this is a relatively small sample size for inference on ring widths.**

Response: We thank the reviewer for this observation. All sampled trees (20 per health category) were included in the cross-dating and ring-width analyses. For the stable carbon isotope analysis, we selected the four best correlated trees per category following recommendation by McCarroll and Loader (2004) for dendro-isotopic studies. We agree that this methodological detail should be clarified, and we will revise the manuscript accordingly to ensure clarity regarding sample sizes used for each type of analysis.

**L159, 162. What were the criteria used for validation? It would be good to at least report sensitivity and perhaps the relative frequency of missing rings in later years, particularly for dying trees.**

Response: The cross-dating of the individual tree-ring series were validated using the COFECHA and ARSTAN programs. No missing rings were detected in any of the samples, including those from dead trees.

Chronology statistics for each health category were as follows:

Healthy trees: mean sensitivity = 0.207, Rbar = 0.195, EPS = 0.938

Damaged trees: mean sensitivity = 0.232, Rbar = 0.198, EPS = 0.936

Dead trees: mean sensitivity = 0.229, Rbar = 0.153, EPS = 0.918

All EPS values exceeded the commonly accepted threshold of 0.85. Mean sensitivity values were within the expected range for natural forests, reflecting moderate interannual variability in growth.

**L165. Why would pre-whitening correct climate-driven growth anomalies? And, what is the benefit of doing so given the goal is (I think) to understand the effect of extreme climate on tree growth?**

Response: Thank you for this important observation. We acknowledge that the original sentence was misleading and did not accurately reflect the purpose of pre-whitening. Given that one of the goals of this study is to identify interannual stress signals—such as those associated with pest outbreaks or climate extremes—pre-whitening is useful for removing low-frequency trends and enhancing the detection of high-frequency anomalies. This facilitates the identification of specific stress events that may not be apparent in the raw data. We will revise Section 2.3 to clearly explain the chronology development process and the rationale for applying pre-whitening in this context.

**L169. Do the authors mean «four trees»? Were cores intended for isotopic analysis surfaced in any way prior to sectioning?**

Response: Yes, by «four cores» we refer to four individual trees per health category. The cores selected for isotopic analysis were not surfaced, as they were kept intact to preserve the wood structure and prevent contamination prior to sectioning for stable carbon isotope measurements. We will revise the manuscript to clarify both points.

**L177. Why not use individual rings for the same sets of years across health statuses?**

Response: We appreciate the reviewer's suggestion, and agree that using individual rings for the same sets of years across health categories would be an ideal approach to improve comparability. Instead, during the isotope analysis, we followed the procedure outlined below:

- For dead trees, we established 2016 as the final growth year (the last ring formed before death), based on field observations and confirmation from local forestry authorities. From this reference, we traced rings backwards to 1993. Individual rings were preserved for isotope analysis for the years 2016, 2011, 2006, 2001, and 1996.

- For healthy and damaged trees, which were still alive at sampling in 2022, we applied a similar strategy: individual rings were preserved for the years 2022, 2017, 2012, 2007, 2002, and 1997.

While this method did not produce samples from exactly the same calendar years across tree health status, we consider this discrepancy minor and not critical for the objectives of the study. Nevertheless, we will clearly acknowledge this limitation in the revised manuscript and discuss its potential implications for interpreting our results.

**L206. H6 in Fig 3 appears to also show progressive damage. This raises the question for me of how damage/health categories were assigned, as this is not described in L146.**

Response: Thank you for your comment. We agree that tree H6 in Fig. 3 appears to show signs of progressive defoliation, particularly in the 2022 image. However, we classified H6 as a healthy tree based on field-based observations at the time of sampling. From the ground level, no visible signs of crown dieback or significant defoliation were observed.

We apologize for not clearly describing our classification criteria in the original manuscript. In this study, we followed the classification scheme of Leidemer (2025), which defines six tree health classes based on the level of defoliation observed from UAV images: healthy (0% defoliation), light damage (≤20%), moderate damage (21–50%), severe damage (>50%), dead (100% defoliation), and fallen (tree on the ground). Based on this scale and field level inspection, H6 fell within the «healthy» category. Furthermore, we recognize that crown appearance may vary across

years in aerial images due to seasonal conditions (e.g., snow cover, lighting, canopy angle), which can complicate interpretation. We will revise the manuscript to clarify our classification methodology and acknowledge this potential source of uncertainty.

**L219. No statistical support in this results section?**

Response: We appreciate the reviewer's comment regarding the lack of sufficient statistical support in the Results section. As noted above, we will expand descriptive information of the sampling sites to include key parameters with associated statistics and relationships between health categories. Additionally, we will conduct Basal Area Increment (BAI) comparisons across health categories and perform correlation analyses between climatic variables and tree-ring and $\Delta^{13}C$ chronologies. We believe that these amendments will enhance the analytical clarity and scientific rigor to the manuscript.

**L225-227. I would suggest to report a correlation**

Response: Thank you for your suggestion. We apologize for not reporting the correlation in the original manuscript. In the revised version, we will include the correlation coefficient to support the relationship described and ensure the results are presented with appropriate statistical backing.

**L230. How much higher? Please report? How is the climate different?**

Response: Thank you for your question. The dead trees used in this study were located at higher elevations, ranging from 1600 to 1714 m a.s.l., near the treeline. In contrast, the healthy and damaged trees were sampled at lower elevations, between 1468 and 1535 m a.s.l. This altitudinal difference corresponds to markedly distinct environmental conditions. The upper plots experience lower mean annual and growing season temperatures, stronger and more persistent winds, greater snow accumulation and longer snow-melting periods. We will include this information in the revised manuscript to clarify these differences.

**L232. So, some of the trees did not die in 2016? How can the authors be confident all the dead trees they sampled died in the same year, given they were sampled 6 years after mortality?**

Response: Thank you for this important comment. While we cannot confirm with absolute certainty that all dead trees died in the same year, we took several steps to ensure consistency in our sampling. According to local forestry officials and the direct observations by ropeway personnel who monitor the mountain regularly, widespread tree mortality in the study area occurred around 2016. In the field, we selected dead trees that showed similar levels of decay and bark retention, indicating a comparable time since death. During tree-ring analysis, we also

verified the final year of growth through visual cross-dating, and in all sampled dead trees the last visible ring corresponded to 2016. While there is always some uncertainty when sampling several years after mortality, we are confident that the sampled trees died within a narrow time window around 2016. We will clarify this point and explicitly acknowledge this limitation in the revised manuscript.

**L240 no uncertainty is reported.**

Response: Thank you for this observation. In dendrochronological studies, the reliability of a chronology is typically assessed using the Expressed Population Signal (EPS), with values above 0.85 indicating a strong common growth signal at the stand level. In our study, all chronologies exceeded this threshold, with EPS values of 0.94 for healthy trees, 0.94 for damaged trees and 0.92 for dead trees, demonstrating high internal coherency and robustness of the resultant chronologies. We will include the EPS values in the revised manuscript to clearly convey the statistical confidence associated with the developed chronologies.

**L242. I would suggest increasing trends are present in a and b. How were trends quantified?**

Response: Thank you for your suggestion. In the original version of the manuscript, the interpretation of $\Delta^{13}C$ data was based solely on visual assessment. We acknowledge that no formal trend analysis was conducted, and we apologize for this oversight. In the revised manuscript, we will re-examine the data and quantify trends using linear regression. If significant trends are identified, we will report them explicitly and revise the relevant figure captions and text accordingly.

---

## Author Comment (AC2)

**Responses to Reviewer #2: RC2**

We sincerely appreciate your thorough evaluation and thoughtful feedback on our manuscript. Below, you can find the answers and explanation to all points you raised. All comments were incorporated in the revised manuscript.

On behalf of all co-authors,

Anna Trigubenko

**General comments:**

**Reviewer#2: Statistical analysis are missing to support their findings**

Response: We appreciate the reviewer's comment regarding the need for stronger statistical support. In the revised manuscript, we will include additional analyses to address this concern. Specifically, we will provide detailed geographical, structural and dendrochronological characteristics of the sampled sites, categorized by tree health status. We will also incorporate Basal Area Increment (BAI) calculations to better illustrate growth differences among healthy, damaged, and dead trees. Furthermore, we will present growth-climate and $\Delta^{13}$C-climate analyses, even in cases where correlations appear weak, in order to illustrate the potential influence of climate on tree physiology.

**Reviewer#2: Methodology is lacking details:**

**- Both site descriptions are missing: for example, tree height, species, age, beetle infestation…**

Response: Thank you for your comment. We agree this information was not sufficiently detailed or clearly structured in the original manuscript. In the revised version, we will expand the description of the study site. Additionally, we will include descriptive tables summarizing key stand and dendrochronological characteristics for each site to improve readability.

**- No clear characterization for healthy, damaged and dead trees. I assume the identification was done visually. What are the exact criteria**

Response: We thank the reviewer for the important comment. We apologize for the lack of clarity regarding the tree health classification criteria. The classification for the Zao site was based on the categories proposed by Leidemer et al. (2025), who defined six distinct tree health classes. These were: healthy (no visible defoliation), light Damage (up to 20% defoliation), medium damage (21-50% defoliation), heavy damage (over 50% defoliation), dead (100% defoliation), and fallen (tree

lying on the ground). The level of defoliation is based on the proportion of white pixels to the total number of pixels within a tree canopy in images taken by Unmanned Aerial Vehicles (UAVs) In order to enhance the performance of the YOLO (deep learning model) classification model, two additional dummy categories were included: artificial object and human. The initial categorization was indeed performed visually, using high-resolution aerial imagery and field validation as supporting references.

**- How many cores were used for tree ring data**

Response: Two cores were extracted per tree. One of the two cores was used for tree-ring width analysis and the other one for stable carbon analysis. In total, 20 trees for each health category were used for tree-ring width analysis.

**Reviewer#2: No clear link between climate factors and beetle infestation is visible, as no statistical analysis were included**

Response: We acknowledge the importance of exploring climatic influences on beetle-related tree decline. To address this, we will conduct a correlation analysis between monthly mean air temperatures during the growing season (April to September) and tree-ring widths for each health category. The analysis covered the period from 1993 to 2022 for healthy and damaged trees, and from 1993 to 2016 for dead trees (due to the tree death dates). The results revealed very weak correlations across all categories. For example, healthy trees showed the highest $R^2$ in April (0.08), while damaged and dead trees exhibited even lower values. The only relatively notable correlation was observed in May for dead trees ($R^2 = 0.16$), though still not statistically significant.

Given the consistently poor correlation results, we decided not to include these data in the manuscript. However, we recognize their relevance in supporting the conclusion that temperature alone is not a sufficient explanatory factor for beetle infestation dynamics.

**Line comments:**

**L20-22. Please check sentence structure**

Response: We will modify the sentences for better clarity as follows:

At the treeline of the Zao Mountains in northeastern Japan, a dual pest outbreak involving the tortrix moth (*Epinotia piceae*) and bark beetles (*Polygraphus proximus*) has caused severe mortality events in natural *Abies mariesii* forests. This is the first reported case worldwide of treeline retreat caused by bark beetle infestation.

**L36. Replace with «these events»**

Response: Done.

**L60-69. Please add references**

Response: Thank you for your comment. We will add appropriate references to support the paragraph. The revised text will read as follow:

As with other bark beetles, once *Polygraphus proximus* populations reach outbreak levels, they are capable of infesting even healthy trees. Such outbreaks, similar to those caused by the mountain pine beetle (*Dendroctonus ponderosae*), have led to the mortality of seemingly healthy trees across millions of hectares (Raffa et al., 2008). In the Zao Mountains, a large-scale bark beetle outbreak between 2012 and 2016 resulted in the devastation of pristine *Abies mariesii* forests over hundreds of hectares, especially those close to the treeline. This outbreak has drastically altered the landscape and is expected to have long-term ecological consequences in the region (Takagi, 2022). Bark beetle infestations not only reduce timber production and quality but also disrupt nutrient cycling, carbon uptake, and ecosystem biodiversity (Jönsson et al., 2009), highlighting the far-reaching impacts of these disturbances. Bark beetle-induced tree mortality also reduces the recreational and economic value of forests, affecting human health, tourism, and local livelihoods (Seidl et al., 2011).

References:

Raffa, K.F., Aukema, B.H., Bentz, B.J., et al.: Cross-scale drivers of natural disturbances prone to anthropogenic amplification: the dynamics of bark beetle eruptions. BioScience, 58(6), 501-517, https://doi.org/10.1641/B580607, 2008.

Takagi, E.: Host preference of the tree-killing bark beetle *Polygraphus proximus* across a geographic boundary separating host species, Entomologia Experimentalis et Applicata 170(11), https://doi.org/10.1111/eea.13229, 2022.

Jönsson, A.M., Appelberg, G., Harding, S., & Bärring, L.: Spatio-temporal impact of climate change on the activity and voltinism of the spruce bark beetle, *Ips typographus*. Global Change Biology, 15(2), 486-499, https://doi.org/10.1111/j.1365-2486.2008.01742.x, 2009.

Seidl, R., Schelhaas, M.J., & Lexer, M.J.: Unraveling the drivers of intensifying forest disturbance regimes in Europe. Global Change Biology, 20(9), 2785-2799, https://doi.org/10.1111/j.1365-2486.2011.02452.x, 2011.

**L71. Which area is meant with «affected area». Zao mountains?**

Response: Thank you for your comment. By "affected area" we refer specifically to the portion of the Zao Mountains that has been impacted by bark beetle infestation. We will clarify this terminology in the revised manuscript to avoid any ambiguity.

**L81-83. Please check sentence structure**

Response: Thank you for pointing this out. We will revise the sentence for clarity. The revised sentence will read as follows:

In addition to tree-ring width measurements, stable carbon isotope analysis of individual rings is a powerful method for assessing environmental impacts on tree physiology.

**L129. Please add location of meteorological station. It would also be beneficial to mark where trees were sampled**

Response: We have revised the manuscript to specify that climate data were obtained from the Zao Ropeway meteorological station, located at 1,661 m a.s.l. (coordinates approx. N 38° 09′, E 140° 26′), adjacent to the upper study site.

To improve spatial clarity, we have added a schematic site map indicating the distribution of sampling plots along the elevational gradient in the Zao Mountains. Furthermore, in the revised version of the manuscript, we plan to include more detailed geolocation data for each sampled tree by health category. We believe these additions will significantly enhance the transparency and reproducibility of our study.

[Figure]

**L138-141. Does not belong in site description**

Response: We agree that this sentence, which outlines the integrative approach and broader objectives of the study, is more appropriate in the Introduction rather than in the Methodology. We will amend the text accordingly.

**L144. From which direction were core taken and/or direction of the slope?**

Response: Thank you for your question. Tree cores were taken from the west-facing side of the trunk, maintaining a consistent orientation across all sampled trees. Coring was done perpendicular to the slope direction to minimize the impact of reaction wood on the tree-ring patterns. We will add this information to the revised manuscript.

**L229. 4 trees per category were used for the TRI?**

Response: No, twenty individual trees per health category were used for the TRI analysis. Each tree was represented by a single core, which is a common and widely accepted practice in dendrochronology when the sample quality is high and cross-dating is robust. The selected trees had well-preserved, complete cores, allowing reliable cross-dating and subsequent TRI calculation. We will revise the manuscript to clearly state the number of trees used per category for the TRI analysis.

**L230. Please add the site description**

Response: We will add a detailed site description to clarify the context. The dead trees used in this study were located at higher elevations, ranging from 1600 to 1714 m a.s.l., near the treeline. In contrast, the healthy and damaged trees were sampled at lower elevations, between 1468 and 1535 m a.s.l. This altitudinal difference corresponds to markedly distinct environmental conditions. The upper plots experience lower mean annual and growing season temperatures, stronger and more persistent winds, greater snow accumulation and longer snow-melting periods.

**L232. It is assumed that all trees died in that year due beetle attack? How was the verified?**

Response: Yes, it is assumed that all sampled trees died around 2016 due to the bark beetle outbreak. This was verified through consistent final growth rings dated to 2016 in all dead trees, based on visual cross-dating. Field selection focused on trees with similar decay stages and bark retention. Additionally, $\delta^{13}C$ values showed a marked decline starting in 2012 and remained low until 2016, confirming a physiological response consistent with infestation and supporting the estimated year of death. We will clarify this in the revised manuscript.

**L240. Sample number is missing**

Response: Thank you for your suggestion. While the total number of sampled trees (20 trees per health category) was described in the Methodology section, we agree that explicitly indicating the sample size in the figure would improve clarity. We will amend the figure accordingly.

**L263. Sample number is missing**

Response: Thank you for your comment. The figure presents $\Delta^{13}C$ records based on a subset of four selected trees pooled together for isotope analysis, with individual rings analyzed every five years. We will include this information to the figure caption to improve its readability.

**L280. What analysis were done to prove «significant»**

Response: This sentence will be removed.

**L301: Is it the average annual wind direction distribution or for a season?**

Response: Thank you for your question. The wind direction distribution described in the manuscript reflects the annual average wind direction distribution in the area.

**L301. Are these the findings of the authors or from previous publications?**

Response: The wind data were obtained directly from the Zao Ropeway Company, which operates weather monitoring equipment near Jizo Mountain. This facility provides continuous in situ measurements, which were analyzed by the authors specifically for this study and have not been previously published.

**L322. Sentence implies that tree mortality was only due to climate**

Response: We will modify the sentence to avoid any ambiguity.

**L326. Please check sentence structure**

Response: Thank you for pointing this out. We will revise the sentence to improve its readability as follows:

Thus, the case of Zao Mountain represents a rare instance of treeline recession. According to a meta-analysis by Harsch et al. (2009), only 1% of global treelines have shown a downward shift, while 52% have advanced upwards.

**L329. Please add the site description**

Response: Thank you for the comment. We will revise the manuscript to include the following site description:

The study was conducted at two distinct elevations in Zao Mountains. The lower elevation site (1468–1535 m a.s.l.) comprises relatively dense stands of *Abies mariesii*, occasionally mixed with other species, and includes the healthy and damaged trees. The higher elevation site (1600–1714 m a.s.l.) is located near the treeline and is composed almost entirely of *Abies mariesii*. This upper site includes the dead trees and is more severely impacted by the bark beetle outbreak. The two sites differ markedly in environmental conditions: the upper plots experience lower mean annual and growing season temperatures, stronger and more persistent winds, greater snow accumulation, and longer snow-melting periods. The average tree age across both sites ranges from 40 to 90 years.

**L357. What analysis were done to prove «significant»**

Response: This sentence will be changed to «The observed patterns confirm that dead trees experienced a sharp reduction in carbon assimilation before their demise, while healthy trees maintained stable physiological function.

**L360. What would be a critical level. Please elaborate**

Response: Thank you for the comment. In our study, we did not define a specific numerical threshold of $\Delta^{13}C$ as a «critical level» associated with mortality. We used the term «critical» in a qualitative sense, referring to sustained reduction in carbon assimilation that may compromise a tree ability to meet metabolic demands and initiate recovery. To avoid confusion, we will revise the sentence. The revised version will read:

Damaged trees showed some fluctuations in $\Delta^{13}C$ values, indicating physiological stress, although not yet reaching levels leading to terminal decline or mortality.